# Endocrine Disrupting Toxicity of Bisphenol A and Its Analogs: Implications in the Neuro-Immune Milieu

**DOI:** 10.3390/jox15010013

**Published:** 2025-01-17

**Authors:** Erica Buoso, Mirco Masi, Roberta Valeria Limosani, Chiara Oliviero, Sabrina Saeed, Martina Iulini, Francesca Carlotta Passoni, Marco Racchi, Emanuela Corsini

**Affiliations:** 1Department of Drug Sciences, Pharmacology Section, University of Pavia, Via Taramelli 12/14, 27100 Pavia, Italy; roberta.limosani01@universitadipavia.it (R.V.L.); chiara.oliviero01@universitadipavia.it (C.O.); sabrina.saeed01@universitadipavia.it (S.S.); racchi@unipv.it (M.R.); 2Department of Pharmacology, Physiology & Biophysics, Chobanian & Avedisian School of Medicine, Boston University, Boston, MA 02215, USA; 3Computational and Chemical Biology, Italian Institute of Technology, Via Morego 30, 16163 Genova, Italy; mirco.masi@iit.it; 4Laboratory of Toxicology, Department of Pharmacological and Biomolecular Science, University of Milan, Via Balzaretti 9, 20133 Milan, Italy; martina.iulini@unimi.it (M.I.); francesca.passoni@unimi.it (F.C.P.); emanuela.corsini@unimi.it (E.C.)

**Keywords:** EDC, RACK1, in vitro screening tool, immune system, INEN, system toxicology, HPA axis, glucocorticoids, BDNF, neurodegeneration

## Abstract

Endocrine-disrupting chemicals (EDCs) are natural or synthetic substances that are able to interfere with hormonal systems and alter their physiological signaling. EDCs have been recognized as a public health issue due to their widespread use, environmental persistence and the potential levels of long-term exposure with implications in multiple pathological conditions. Their reported adverse effects pose critical concerns about their use, warranting their strict regulation. This is the case of bisphenol A (BPA), a well-known EDC whose tolerable daily intake (TDI) was re-evaluated in 2023 by the European Food Safety Authority (EFSA), and the immune system has been identified as the most sensitive to BPA exposure. Increasing scientific evidence indicates that EDCs can interfere with several hormone receptors, pathways and interacting proteins, resulting in a complex, cell context-dependent response that may differ among tissues. In this regard, the neuronal and immune systems are important targets of hormonal signaling and are now emerging as critical players in endocrine disruption. Here, we use BPA and its analogs as proof-of-concept EDCs to address their detrimental effects on the immune and nervous systems and to highlight complex interrelationships within the immune–neuroendocrine network (INEN). Finally, we propose that Receptor for Activated C Kinase 1 (RACK1), an important target for EDCs and a valuable screening tool, could serve as a central hub in our toxicology model to explain bisphenol-mediated adverse effects on the INEN.

## 1. Introduction

Endocrine-disrupting chemicals (EDCs) are natural or synthetic substances that may mimic, antagonize or interfere with the body’s hormones and, thus, the endocrine system, representing a recognized public health issue [1]. According to the European Commission, an endocrine disruptor is defined as a substance able to cause adverse effects either in an organism or its progeny or (sub)populations. These adverse effects, which include alterations in the morphology, physiology, growth, development, reproduction or lifespan, can lead to negative changes in functional capacity, in their stress-compensating ability or to an increased sensitivity to other influences. Furthermore, a substance is classified as an EDC when it has the ability to change the functions of the endocrine system, and, therefore, its adverse effects are a consequence of the disruption of endocrine functions [1]. There is a growing body of evidence suggesting the existence of intricate interactions among the endocrine, immune and nervous systems, referred to as the immune–neuroendocrine network (INEN) [2,3]. Indeed, the physiological mechanisms that control the INEN’s activity include hormones, cytokines, neurotransmitters and neuropeptides, as well as their interactions with each other, which influence various physiological processes [3]. Hence, the aim of this review is to underscore the effect of bisphenol A (BPA) and its widely used analogs, BPAF, BPF and BPS, in this intricate communication network, which can vary based on the cellular context (i.e., receptors and interacting proteins) and may therefore differ between tissues and conditions. The resulting multifaceted response of BPA and its analogs is also investigated through the lens of the scaffold protein Receptor for Activated C Kinase 1 (RACK1), an EDC target, shown to be regulated by a complex hormonal balance (reviewed in [4,5]) and, thus, representing a plausible tool to unraveling the interplay between immune and nervous systems.

## 2. BPA and Its Analogs BPAF, BPF and BPS

### 2.1. Key Features of BPA and Its Analogs BPAF, BPF and BPS

BPA, also known by the IUPAC name as 4-[2-(4-hydroxyphenyl) propane-2yl] phenol, was first synthesized in 1891 by Russian chemist AP Dianin [6]. BPA is a solid, white and crystalline substance that is highly soluble in fats and with low water solubility. BPA is a widespread EDC, classified by the Classification, Labeling, and Packaging (CLP) Regulation ((EC) No. 1272/2008) as potentially harmful to reproduction, as well as the skin, respiratory tract and eyes, affecting multiple physiological receptors with different agonistic or antagonistic effects depending on the target tissue and cell type [7,8]. BPA can bind to the nuclear estrogen receptor subtypes (ERα and ERβ) [9,10,11] as well as to the membrane G protein-coupled estrogen receptor (GPER, also known as GPR30) [9,12], which can induce rapid non-genomic estrogenic responses [13]. Due to its chemical structure, BPA can also bind to estrogen-related receptor γ (ERRγ) [9,10,14], androgen receptor (AR) [9,10], thyroid hormone receptor subtypes (THRα and THRβ) [9,10], glucocorticoid receptor (GR) [10,12], progesterone receptor (PR) [10] and peroxisome proliferator-activated receptor γ (PPARγ) [9,10]. Hence, the toxicity of BPA and its analogs is related to their interaction with hormone receptors on specific cell types whose expression is modulated according to age, sex, endocrine status and brain region. Indeed, this highlights that exposure to bisphenols could be associated with different adverse health effects in humans (reviewed in [15,16]). Indeed, harmful impacts on different systems, including the nervous, endocrine and immune systems, were reported (reviewed in [17]). The primary routes for bisphenol exposure in humans are occupational, environmental and contaminated food consumption. Occupational exposure occurs in workers involved in the synthesis of BPA and its derivatives (i.e., polyvinyl chloride, polycarbonate and epoxy resins), but also in cashiers exposed to BPA via dermal penetration through thermal receipts [18], and other workers (e.g., employees of sewage-pipe relining companies and floor-coating companies) [19]. On the other hand, environmental exposure—also as a consequence of its employment in thermal paper recycling and related industrial processes—results in aquatic system, atmosphere and soil contamination [20]. Moreover, it is extensively used as a primary ingredient for the production of plastics employed in agricultural, industrial and medical applications. It is also an epoxy resin key component employed for coatings in canned food and beverages, polycarbonate plastic bottles, food containers, dental sealants, water supply pipes, toys and cigarette filters [21,22]. Therefore, BPA human exposure occurs through both dietary sources (food packaging, liquid containers and microwave utensils) and non-dietary sources (electronic and sports equipment, thermal paper, paints and printing inks, flame retardants, medical components and DVDs), which contaminate air, water, soil, landfills and food, affecting humans directly or indirectly through various exposure routes (reviewed in [23]). Dietary oral exposure is the primary route, which includes consumption of BPA-contaminated freshwater fish or seafood, polluted regions’ fresh food and ingestion of plastic/can-packaged food and polluted water [24]. In addition, breast milk, cow milk and dairy products represent additional notable sources for BPA oral exposure [25]. BPA absorption through dermal exposure is the second most important route [26]. Direct contact with paper (particularly thermal paper), medical devices and toys proportionally increases potential BPA exposure to the skin. Exposure through inhalation is the third main route, occurring via BPA-containing vapors, mists, dust and gases [24]. Therefore, BPA exposure levels are highly dependent on the proximity to an industrial site, lifestyle and diet [27,28]. The European Union Food Safety Authority (EFSA) set a tolerable daily intake (TDI)—defined as the threshold for daily human exposure to a chemical pollutant considered safe for consumer health—for BPA of 0.2 ng/kg body weight (bw)/day for human health over a lifetime of chronic exposure [29]. This is a dramatic reduction from the 4 μg of BPA/kg bw/day established in 2015 [30]. However, the TDI is not completely reliable and safe since BPA exhibits a non-monotonic dose–response curve, showing that even very low doses of exposure pose a risk to public health [31]. This has led to the development of structurally similar BPA substitutes (Figure 1), of which BPAF, BPF and BPS are the most widely used in household products as a raw material for epoxy and polycarbonate resin, thermal papers and plastic reinforcement [28].

BPA is highly metabolized and excreted in the urine within 24 h, mainly as a glucuronide conjugate and, to a lesser extent, as a sulfate conjugate, and these metabolites are commonly used for biomonitoring studies [32,33,34]. A physiologically based pharmacokinetic (PBPK) model for oral and dermal exposure to environmentally relevant concentration ranges to BPA was used to assess the hepatic and intestinal glucuronidation kinetics of BPAF, BPF and BPS [35,36,37,38]. The highest internal concentrations of unconjugated bisphenols resulted in both the serum and the gonads, with oral exposure being the most relevant route for BPS and dermal exposure for BPAF [35]. In addition, human biomonitoring data have demonstrated that bisphenols are widespread, and their urinary concentrations are increasing, thus highlighting continuous human exposure [36,37,38]. BPA is fat-soluble, and consequently, it has a high affinity for adipose tissue, where it has been found with a 1.13–12.27 ng/g range, although it could also be found in other tissues in humans and mice [39,40,41], including a 1–2.35 ng/g range observed in the brain [42]. Indeed, BPA accumulation was reported in post-mortem human brains [24]. However, the non-excreted fraction can be detected at lower concentrations in biological fluids such as blood, amniotic fluid, placental tissue, umbilical cord blood, breast milk and human colostrum, thus also highlighting adverse health effects in the prenatal phase [43,44,45,46]. The literature data report that BPAF, BPF and BPS at concentrations of 0.001–1 µM exert estrogenic effects. In this regard, BPAF exerts the strongest effect as both agonist and antagonist [47,48]. Moreover, Rosenmai and collaborators showed that BPA and its analogs, more specifically BPF and BPS, also exerted anti-androgenic effects [49]. Therefore, the disruptive effects of BPA analogs were similar or greater than those of BPA due to their interaction with these hormone receptors in a similar way to BPA [28,50,51,52]. Hence, they do not appear to reduce the risk of endocrine disruption and need to be carefully considered as alternatives to BPA [35].

### 2.2. An Overview of the Health Effects of BPA and Its Analogs

#### 2.2.1. Effects of BPA and Its Analogs on Neurodevelopment and Behavior

Epidemiological studies suggest that BPA prenatal exposure could alter the brain’s structure and function [53,54,55,56,57,58], leading to neurodevelopmental disorders, such as learning disabilities, sensory deficits, development delays, attention deficit hyperactivity disorder (ADHD) and autism spectrum disorder (ASD) (reviewed in [59,60]). Recently, the associations between BPF or BPS with ADHD symptoms have been analyzed at multiple time points in children (at ages 4, 6 and 8), reporting that all bisphenols were associated with ADHD symptoms at age 6 [61]. Moreover, recent in vivo animal data suggest that BPF and BPS, at sublethal concentrations (1, 10 or 100% of their respective EC_25_), may also pose significant risks to ASD development in humans, highlighting the importance of a comprehensive assessment of developmental neurotoxicity effects for bisphenols [62,63]. Indeed, the disruptive effects of bisphenols on neurodevelopment have been examined mainly in mammals but also in other species, while the molecular mechanisms have been investigated in more detail in various in vitro studies (reviewed in [59]). Epidemiological studies on emotional neurobehavioral problems showed an association between parental urinary BPA concentrations and depressive, anxious and hyperactive behaviors in children (Table 1) (reviewed in [59]), although a few studies found no association [64,65,66]. On the other hand, there are limited data on BPA alternatives, suggesting potential neurodevelopmental risks, although specific analogs (BPAP, TBBPA) induced hippocampal changes that are linked to recognition behavior, sociability and increased anxiety levels, while other analogs (BPS, BPF, BPE, BPB, BPZ and BPAF) exhibited neurotoxic effects in vitro, suggesting neuroinflammatory and microglial activation [67].

Recently, BPA’s and BPS’s potential influence on child behavior at 2 years was assessed using the child behavior checklist (CBCL). Median concentrations for BPA and BPS were 1.3 ng/mL (range 0.4–7.2) and 0.3 ng/mL (range 0.1–3.5), respectively. Noteworthily, the relationship between BPA or BPS and the scores on several syndrome scales (aggressive, anxious–depressed and sleep problems) was significantly different between the sexes. Higher BPA or BPS concentrations correlated with higher levels of problematic internalizing behaviors in girls but not boys, thus suggesting sex-specific associations between urinary bisphenol’s concentration and problematic child behaviors [77]. However, BPA, BPF and BPS effects on depressive symptoms were also evaluated in adults. No significant correlation was found between the BPA urine levels and depressive symptoms in the general population. In stratified analyses, urinary BPA and BPS were positively associated with depressive symptoms in men. Indeed, in elderly men (≥60 years old), urinary BPA and BPS were positively correlated with depressive symptoms. Moreover, BPS was negatively associated with depressive symptoms in elderly women (≥60 years old) [78]. Altogether, these findings highlight the impact of bisphenols on the stress-response system and, consequently, on the hypothalamic–pituitary–adrenal (HPA) axis, as supported by the Alberta Pregnancy Outcomes and Nutrition (APrON) cohort study, which provided the first human evidence that BPA exposure during pregnancy is associated with dysregulation of the HPA axis’ function [79]. The HPA axis is the neuroendocrine effector of the system that controls reactions to stress through a complex set of direct influences and feedback interactions among the hypothalamus, the pituitary gland and the adrenal glands. Extra-hypothalamic brain areas (i.e., the amygdala, hippocampus and prefrontal cortex) mutually interact with the HPA axis in both basal and under stress conditions. Following stress, paraventricular hypothalamic neurons release corticotrophin-releasing hormone (CRH), arginine vasopressin (AVP) and oxytocin (OT), which trigger the secretion of adrenocorticotropic hormone (ACTH) from the anterior pituitary. In turn, ACTH stimulates the release of glucocorticoids (GCs) (mainly cortisol in humans) from the adrenal cortex. Circulating GCs act on peripheral tissues by binding to their receptor to produce a wide range of effects and act back on the hypothalamus and pituitary to suppress CRH and ACTH production via a negative feedback loop [80]. While short-term elevations in GCs are beneficial for physiological function and survival, long-term exposure to GCs leads to the suppression of immune and neural functions through inhibition of neuronal and glial resilience, glucose uptake and energy balance, resulting in neurotoxicity [81,82]. In addition, chronic exposure to elevated GCs can result in altered social, anxiety, depressive and reproductive behaviors. In this regard, BPA has been reported to interfere with hormonal balance by disrupting estrogen and androgen receptor function, leading to altered levels of follicle-stimulating hormone (FSH) and luteinizing hormone (LH). This imbalance can contribute to reproductive disorders, including polycystic ovary syndrome (Figure 2). Finally, other downstream physiological consequences also include metabolic and cardiovascular diseases [83,84,85,86].

The molecular mechanisms involved in the neurotoxic effect of BPA have been investigated in animal studies, primarily reporting that an abnormal development of the HPA axis can lead to long-term changes in the synthesis of neuropeptides and neurotransmitters in the central nervous system (CNS), as well as in the synthesis of glucocorticoid hormones in the periphery. This potentially leads to the observed disruption of neuroendocrine, behavioral and metabolic functions in adulthood (reviewed in [16,59,60,87]). In this regard, a part of the CLARITY-BPA consortium study reports that exposure to different BPA concentrations (2.5, 25, 250, 2500 or 25,000 μg/kg bw/day) alters the transcriptome of the neonate rat amygdala in a sex-specific manner. This study also shows that in the developing brain, exposure to BPA can disrupt the estrogen, oxytocin and vasopressin signaling pathways, which are critical for synaptic organization and transmission [88]. In addition, BPA can alter the neurological system from molecular and structural points of view in specific brain regions, mainly in the cortex and hippocampus, thus affecting brain plasticity and suggesting BPA effects on learning and memory. Indeed, adverse effects on neurotransmitters, disruption of intracellular calcium homeostasis and changes in the expression and activity of key genes and proteins, including through epigenetic mechanisms, excitotoxicity, oxidative stress, neuronal damage, neuroinflammation and impaired blood–brain barrier function, have been reported in various in vivo and in vitro models (reviewed in [16,59,87]).

#### 2.2.2. BPA and Its Analogs in Neurodegenerative Disorders

Neurodegenerative diseases are pathological conditions that feature unrestrained neuronal death in different brain regions due to alterations in various cellular and molecular pathways, including protein misfolding, mitochondrial dysfunction, oxidative stress and neuroinflammation, which have also been detected following bisphenol exposure (reviewed in [16,59,89]). In addition, BPA and its analogs interfere with ER and AR, indicating that hormonal changes due to bisphenol exposure could have detrimental effects on the neuroendocrine system and favor neurodegenerative disease development [16]. In this regard, there is some recent evidence of a plausible link between exposure to BPA and neurodegenerative diseases, principally Alzheimer’s disease (AD) and Parkinson’s disease (PD) (reviewed in [16,89]). Epidemiological studies report that liver and adipose tissue BPA concentrations in AD patients were, respectively, 4.24-fold and 6.76-fold higher than those of the age-matched control group [90]. Moreover, the conjugated BPA plasma concentration was significantly reduced in PD patients, indicating a weakened BPA metabolic effect resulting in unconjugated BPA binding to albumin and protein disulfide isomerase [91]. This affects the conformation of the newly folded protein, leading to α-synuclein increased levels and protein misfolding, the hallmark of PD par excellence [92]. In addition, other prospective cohort studies have reported that BPA exposure is associated with typical features of AD and PD, such as abnormal behavior and cognitive abilities [69,93,94]. However, data from in vivo and in vitro studies provide further support that exposure to BPA or its analogs, BPF and BPS, may be a major factor in the increased risk of these pathologies [89].

AD is characterized by neurofibrillary tangles (NFTs), intracellular aggregates of hyperphosphorylated tau protein (p-tau) within neuronal cells, and senile plaques (SPs), which result from the extracellular deposition of the β-amyloid (Aβ) peptide derived from the processing of amyloid precursor protein (APP). APP is processed by at least two different proteolytic pathways. The non-amyloidogenic pathway involves cleavage by the enzyme α-secretase, which cleaves APP within the Aβ sequence, preventing the formation of Aβ, while the amyloidogenic pathway involves cleavage of APP by β-secretase 1 (BACE1), resulting in the formation of Aβ [95,96]. The literature data report that BPA leads to increased BACE1 enzymatic activity in human AD brain extracts concomitantly with higher levels of Aβ accumulating in the cerebral cortex and in the hippocampus of these patients [97]. The literature data of animals suggest that the BPA-induced disruption of insulin signaling may be involved in induced AD-like neurotoxicity through the signaling pathway mediated by Akt and phosphatidylinositol 3-kinase (PI3K)-dependent mechanisms, resulting in the phosphorylation of Akt and activation of glycogen synthase kinase 3β (GSK3β)—a key player dysregulated in AD [98]—leading to dramatic increases in APP and p-tau (reviewed in [89]). In line with these considerations, another study reports increased levels of p-tau protein following BPA exposure [99]. Consistent with these findings, an association between the disruption of insulin signaling and cognitive impairment in the human brain has also been found [100,101]. Moreover, in vitro studies have shown that an increase in tau phosphorylation and Aβ expression in the hippocampus and cerebral cortex after BPA exposure impairs cells’ calcium (Ca^2+^) buffering capacity [102], disrupting Ca^2+^ homeostasis [99]. This results in oxidative stress in human cortical neurons, potentially indicating BPA as a contributor to neurotoxicity through the upregulation of Ca^2+^ levels and the dysregulation of reactive oxygen species (ROS) scavenging [16]. Disruption of Ca^2+^ signaling plays a pivotal role in neuroinflammation development, which is recognized as the activation of glial cells, significantly involved in AD pathogenesis [103,104]. Indeed, their activation induces the release of pro-inflammatory cytokines, chemokines and prostaglandins [105] that promote apoptosis and cytotoxicity in neurons and ultimately lead to neurodegeneration [106,107]. BPA has been found to upregulate the release of inflammatory cytokines, e.g., tumor necrosis factor α (TNF-α) and interleukin-1β (IL-1β), in the hippocampus while affecting the release of the anti-inflammatory IL-10 cytokine, suggesting that BPA may affect the response to glial cell activation [108]. The literature data show that BPF can also exert significant neurotoxicity in vivo, resulting in microglial and astrocyte activation [109]. In addition, BPS nanomolar concentrations could impair cognitive function by triggering oxidative stress in the hippocampus through estrogenic pathways, similar to low-dose BPA exposure (reviewed in [110]). Nevertheless, the scarcity of experimental evidence on the correlation between BPA analogs and the development of neurodegenerative diseases like AD surely warrants further investigations to address these aspects.

PD is the second most common neurodegenerative disorder after AD [111]. The hallmark of this disease is the loss of dopaminergic neurons in the substantia nigra within the midbrain and the widespread aggregation of α-synuclein, which leads to the onset of prominent motor symptoms (bradykinesia). BPA exposure is associated with neurotoxic effects, including oxidative stress and neuronal damage, which are critical in PD pathogenesis. The binding of unconjugated BPA to proteins can lead to protein misfolding and the formation of neoantigens, which may trigger immune responses that further contribute to neurodegeneration [91,92]. PD early-onset can be caused by pathogenic genes, including PARK7, PARK2 and PINK1, essential for monitoring mitochondrial health. PARK7 encodes for the ubiquitous highly conserved deglycase and ROS sensor DJ-1, which modulates the mitochondrial responses and plays a role in maintaining cellular health under stress conditions. PINK1 accumulates on damaged mitochondria, phosphorylating ubiquitin to signal for their degradation via the autophagic-lysosomal pathway, while PARK2 gene encodes for parkin, a protein crucial for the selective degradation of damaged mitochondria through mitophagy. Therefore, PINK1 and parkin work together to eliminate dysfunctional mitochondria. Disruption of this process can lead to mitochondrial dysfunction and increased oxidative stress, both of which are critical factors in PD pathogenesis. The literature data report that BPA may also interfere with these proteins, exacerbating mitochondrial damage and promoting neurodegeneration [112,113,114,115]. In vivo, long-term exposure to BPA also results in a significant downregulation of tyrosine hydrolase (TH), a catalytic enzyme involved in the biosynthesis of dopamine and essential for dopamine homeostasis [116]. Similarly to BPA, BPF and BPS were also reported to induce PD-like symptoms (reviewed in [16,87,89]). Finally, emerging evidence of neurodegenerative effects induced by bisphenols has also been suggested in amyotrophic lateral sclerosis (ALS) and multiple sclerosis (MS) (reviewed in [16,89]).

#### 2.2.3. Impact of BPA and Its Analogs on the Immune System

The immunomodulatory effects of BPA and its analogs on the innate and adaptive immune systems have been investigated both in vivo and in vitro (reviewed in [117]). In this context, despite numerous original studies and reviews on BPA immunotoxicity, only a small number of studies investigated the effects of BPA analogs on the immune system. While exposure to specific BPA analogs resulted in either increased inflammation (BPAP, BPE, BPP and BPZ) or increased immunosuppression (TBBPA, TCBPA), these limited but concerning data pinpoint at an immunotoxic potential of BPA alternatives and warrant further research to investigate how these substances may disturb both innate and adaptive immune functions, leading to immune-mediated health issues [67].

##### Innate Immune System

In the innate immune system, tissue-bound (macrophages and dendritic cells) and motile cells (monocytes, neutrophils and eosinophils) have been described as targets of the immunomodulatory effects of BPA and its analogs BPAF, BPF and BPS. In dendritic cells, BPA (50 μM) and BPAF (30 μM) were found to significantly affect the expression of differential cell markers during dendritic cell differentiation. Indeed, BPA and BPAF induced the expression of Dendritic Cell-Specific Intercellular adhesion molecule-3-Grabbing Non-integrin (DC-SIGN), also known as Cluster of Differentiation 209 (CD209), whereas BPAF induced a decrease in the CD1a, CD80 and CD86 markers. Furthermore, the presence of BPA at high concentrations (50 μM) suppressed lipopolysaccharide (LPS)-induced TNF-α production in cultured mouse macrophages [118]. This was consistent with another study conducted in isolated macrophages from mice orally treated with BPA for 4 weeks, which found that further exposure to BPA (10 and 100 μM) in culture suppressed LPS-induced TNF-α secretion [119]. In contrast, exposure to a lower concentration of BPA (10 ng/mL or 43 nM) showed a stimulatory activity on macrophages, resulting in the LPS-induced release of TNF-α and in the upregulation of surface molecules, including CD40, CD80 and MHC-II, which are important for antigen-presentation and co-stimulation [120]. The increase in LPS-induced TNF-α release was also found in macrophages exposed to BPF and BPS, albeit at higher concentrations (10 and 20 μM, respectively). Indeed, BPF and BPS promoted macrophage M1 polarization and the pro-inflammatory state with a relevant increase in the release of cytokines, including IL-1β and IL-6 [121]. Similar results were obtained in a comparative study of the immunomodulatory effects of BPA, BPAF and BPS at concentrations of 0, 0.1, 1, 10 and 100 µM for over 96 h in phorbol 12-myristate 13-acetate (PMA)-differentiated U937 cells, a widely used model for primary human macrophages. At 0.1 µM, BPA and BPAF increased the release of cytokines/chemokines, whereas at higher concentrations, they suppressed their secretion. BPS had minimal effects at low concentrations, but in line with previous data, high concentrations induced the release of pro-inflammatory cytokines [122]. Similarly, high concentrations of BPA and its analogs BPAF and BPS were immunomodulatory in monocytes in vitro, suggesting that bisphenols may contribute to aberrant immune functions, with BPA and BPAF exerting an immunosuppressive effect. On the other hand, BPS makes immune cells more reactive, leading to various autoimmune or allergic disease developments [117,123]. In this regard, BPA and BPAF decreased the LPS-induced expression of surface markers and the release of pro-inflammatory cytokines, whereas BPS increased the LPS-induced expression of CD86 and cytokines [124]. In addition, BPA and BPS, as well as their glucuronidated metabolites, were reported to affect the energetic metabolism and multiple neutrophils’ anti-microbial effects, with potential health implications [125].

##### Adaptive Immune System

The main protagonists of the adaptive immune system are T and B lymphocytes. The former have cytotoxic, effector and regulatory activities, while the latter are responsible for antibody production. These two subsets of adaptive immunity cells have been described as potential targets of EDCs (reviewed in [117]). BPA has been shown to have significant immunomodulatory effects on T lymphocytes, specifically on T-helper (Th) cell differentiation and T-cell activation and signaling, potentially contributing to diverse immune-mediated diseases. The literature data report that in mice exposed during prenatal stages or adulthood, BPA favors Th2 cell development in adulthood and both Th1 and Th2 cells in prenatal stages due to the reduction of the number of regulatory T cells [126]. Moreover, BPA exposure can impair the Th1/Th2 T cells balance and impact the correct Treg function and differentiation. Consequently, an affected Th1/Th2 ratio could lead to altered cytokine release, impairing immune defense against pathogens or contributing to allergic conditions [127]. Noteworthily, low doses of BPA over-activate and influence Th17 cell differentiation, thus shifting the immune response towards a more pro-inflammatory state, ultimately increasing susceptibility to immune-mediated conditions (e.g., inflammation, allergies and autoimmune diseases) and exacerbating autoimmune responses [29]. This evidence identified the immune system as most sensitive to BPA exposure and prompted EFSA to drastically lower the BPA safe exposure levels in the 2023 re-evaluation of BPA’s public health risk, underscoring how even low-level exposure could have long-term health consequences related to inflammation or immune-related diseases and highlighting how these effects are particularly concerning for vulnerable populations (e.g., children and pregnant women), who are more sensitive to BPA-caused immune system disruptions [29] (Figure 3).

In addition to BPA, the effects of exposure to environmentally relevant concentrations (i.e., 0.05 nM) of BPF and BPS were analyzed in mouse spleen T lymphocytes. Exposure to 0.05 nM BPA or BPAF significantly increased the IL-17 levels, whereas BPS had no effect [128]. In contrast, on human T cells, only the highest concentration of bisphenols, representing the supra-environmental level of exposure (50 µM BPA, BPF or BPS), significantly reduced the secretion of IL-17 and IL-22 [128]. A systematic multi-omics analysis has revealed that BPA exposure induced the overactivation of T-cell receptor (TCR) signaling. Additionally, BPA exposure altered the abundance of CD3 activation-related proteins, including a decreased phosphorylation of the T-cell activation ligand LAT and an increased Src phosphorylation [129]. Since LAT is essential for T-cell development, its reduced phosphorylation not only results in the failure of T-cell development but also leads to T-cell hyperactivation and expansion [130]. Moreover, upon BPA exposure, an increased number of early activated CD3^+^ T cells followed by alterations in cytokine expression were correlated with impaired T-cell function and immune response [129]. Regarding their impact on B-lymphocyte function, BPA, BPF and BPS exposure can significantly reduce B-cell survival and viability. Indeed, these bisphenols show cytotoxicity towards human B cells, with BPA being the most toxic. At 100 μM, BPA decreased B-cell viability by around 70%, while BPS and BPF induced a 20% and 30% reduction of cell viability, respectively. BPA treatment increased the percentage of B cells in the apoptotic sub-G0/G1 region to 26%, significantly higher than 10% for BPS and 12% for BPF [131]. These data were in line with the CLARITY-BPA study results, which found that a daily dose of 25 μg BPA/kg bw in rats resulted in reduced splenic B lymphocytes in females after 6 months of exposure [132,133]. Finally, BPA, BPF and BPAF were found to have a negative effect on the cell viability of peripheral blood mononuclear cells (PBMCs) at high concentrations for long-term exposure, in contrast to BPS, which showed only a slight decrease. Exposure to these bisphenols can lead to significant alterations in PBMC morphology as well as increased apoptosis and necrosis [134,135]. In PBMCs activated with PMA and ionomycin, BPA, BPAF and BPS affected PMA/ionomycin-induced T-helper differentiation and cytokine release with gender-related alterations. BPA exposure has been linked to the modulation of cytokine production in PBMCs. Specifically, low doses (0.1–10 nM) of BPA can enhance cell proliferation while simultaneously decreasing the secretion of anti-inflammatory cytokines, particularly in activated PBMCs. In contrast, the production of pro-inflammatory cytokines may be altered, potentially contributing to immune dysregulation and increasing susceptibility to inflammatory and autoimmune diseases [136,137]. Altogether, these data suggest that bisphenols can have an impact on immune cell differentiation and activation, further hinting at their immunotoxic effects.

#### 2.2.4. BPA and Its Analogs on Other Immunoendocrine-Related Diseases

BPA exposure has also been linked to other health outcomes in childhood and adults, including infertility (reviewed in [138,139]), obesity (reviewed in [140,141]), metabolic and cardiovascular diseases (reviewed in [142,143,144]) and hormone-related cancer risks (reviewed in [145]). In this regard, BPA exposure has been described as a further risk factor for fatty liver disease progression, tightly associated with the crosstalk between inflammation, oxidative stress and immune-metabolic impairment associated with obesity in mice [146]. Aside from BPA, various population-based studies revealed that BPA substitutes (BPAF, BPF and BPS) may act as obesogens at the pathophysiological level (reviewed [147]). BPS has been shown to act through mechanisms, including PPARγ activation, potentiation of high-fat diet-induced weight gain and stimulation of adipocyte hypertrophy and fat depot composition, whereas the evidence for BPF and BPAF is more inconclusive (reviewed in [148]). In addition, obesogenic properties of these bisphenols highlight possible links to an increased cardiovascular risk (reviewed in [148]).

## 3. RACK1 Structure, Function and Context-Dependent Transcriptional Regulation

Among the most studied EDCs, insecticidedichlorodiphenyl-trichloroethane (*p*,*p′*-DTT) and the synthetic non-steroidal estrogen diethylstilbestrol (DES) were both central proof-of-concept substances to the discovery of endocrine disruption and were used in the initial investigation to support RACK1 as a potential target of EDCs and their immunotoxicity [5,149,150,151].

### 3.1. RACK1 Structure and Functions

RACK1 is a member of the tryptophan-aspartate (WD) repeat protein family, highly conserved among eukaryotes and involved in several molecular pathways and biological functions. RACK1 displays a seven-bladed β-propeller structure, characterized by repetitive WD40 domains and small structural motifs that fold into four-stranded anti-parallel β-sheets [152]. The human β subunit of the heterotrimeric G protein (Gβ) was the first WD repeat protein to be characterized and shares 42% identity with human RACK1 [153]. WD repeat family proteins lack enzymatic activity and primarily serve as scaffolds for other proteins, facilitating the formation of the protein complexes necessary for regulating the various pathways involved in maintaining cell homeostasis. RACK1 has been observed to have three possible interaction surfaces that allow its interaction with multiple binding partners, including kinases (MAPK, JNK, Src, PKCβ), integrins, phosphatases, ion channels, membrane receptors, G proteins, apoptosis-related molecules and ribosome-associated structural proteins (reviewed in [4,154,155]). The role of RACK1 has been shown to be cell context-dependent, as it differently regulates specific pathways reflecting the cell type and the differential expression of its binding partners [4,5,153]. Accordingly, RACK1 is involved in diverse biological events such as the immune response, neuronal activity, development, epithelial barrier maintenance and cancer development and progression [153,155,156].

### 3.2. RACK1 Role in Immune and Nervous Context and Its Related Transcriptional Regulation

#### 3.2.1. RACK1 in Immune Context

In the immune context, one of the key roles of RACK1 is to act as PKCβII scaffold protein, stabilizing its active conformation and promoting its translocation to its substrates [157], which are essential for immune cell activation, increased CD86 expression, and cytokine release, specifically TNF-α and IL-8 (reviewed in [151]). The role of the RACK1/PKCβII complex was first investigated in immunosenescence, a progressive and degenerative alteration in the immune system that occurs with age. This condition was then linked to a loss of PKC translocation as a consequence of the age-dependent decrease in RACK1 levels (reviewed in [5]). A 50% lower TNF-α release after LPS stimulation was observed in alveolar macrophages from aged rats (compared to young rats) [158] and, similarly, in human monocytes, macrophages and peripheral blood leukocytes. This was ascribed to an impaired PKC translocation due to the age-related reduced RACK1 expression [159,160]. The literature data show that part of this defective signaling in immune cells can be attributed to age-related alterations in the hormonal balance between cortisol and dehydroepiandrosterone (DHEA) [161], which have been demonstrated to be involved in RACK1 transcriptional regulation [4,5].

##### Glucocorticoids-Mediated RACK1 Transcriptional Regulation in Immune Cells

RACK1 is an important target of corticosteroid-induced anti-inflammatory effects due to the presence of a negative Glucocorticoid Response Element (GRE) in its promoter region [162]. The literature data report that, in human monocytes, cortisol induced a significant downregulation of RACK1 promoter luciferase reporter constructs that was mirrored at RACK1 mRNA and protein levels, correlating with a significant reduction in the cytokine released in response to LPS [161]. The cortisol anti-inflammatory effect was confirmed to be dependent on RACK1 modulation using a RACK1 pseudosubstrate, which directly activates PKCβ and mifepristone, a well-known GR antagonist that prevents RACK1 downregulation [163]. Similar results were obtained with other corticosteroids such as betamethasone, budesonide, methylprednisolone, prednisone and prednisolone [164]. Moreover, it was demonstrated that cortisol modulates the exon 9 inclusion in the GR mRNA transcript by upregulating Serine/arginine-Rich Splicing Factor 3 (SRSF3, also known as SRp20) and downregulating SRSF9 (also known as SRp30c), resulting in a significant increase in GRα isoform, the major glucocorticoid mediator, which, in turn, contributes to RACK1 downregulation [165]. Cortisol activity can be counteracted by dehydroepiandrosterone (DHEA), which promotes GR mRNA splicing towards the β isoform (GRβ) by increasing SRSF9 expression. Therefore, DHEA acts as an anti-glucocorticoid by increasing the expression of GRβ, which antagonizes GRα binding to the GRE site in the RACK1 promoter [165]. In line with these considerations, GRβ silencing abolished DHEA-induced RACK1 increase and cytokine release modulation, confirming that the DHEA effect is driven by the modulation of GRβ expression and activity. Moreover, DHEA involvement in GRβ expression was also confirmed by SRSF9 silencing, strengthening their hypothesized correlation. Indeed, SRSF9 knockdown completely blocked a DHEA-induced GRβ increase with the consequent prevention of DHEA-induced RACK1 expression [165,166]. Finally, it was demonstrated that DHEA exerts anti-glucocorticoids properties on RACK1 not only by interfering with GR splicing but also through its conversion in active androgen steroids, indicating that RACK1 regulation is modulated also by androgens [167].

##### Androgen-Mediated RACK1 Transcriptional Regulation in Immune Cells

The literature data obtained both via in vitro and in vivo studies showed that AR and GR can form heterodimers at a common DNA site that comprises a well-conserved inverted repeat of the 5′-AGAACA-3′ hexamer with a 3-nt spacer, i.e., canonical androgen/glucocorticoid response element (ARE/GRE) [168,169,170]. Accordingly, the non-canonical GRE sequence, described on the RACK1 promoter, is also a cis-regulatory target of AR that positively regulates RACK1 expression [4,5,149]. In this regard, the literature data showed that DHEA can be metabolized to androgens, and these androgenic metabolites promote RACK1 expression and, consequently, monocyte activation. Androgen-mediated RACK1 transcriptional regulation was also confirmed by both AR silencing and flutamide, a well-known AR antagonist [161,167]. Finally, ligand-independent AR activation by 17β-estradiol through the PI3K/Akt signaling cascade involving GPER showed an increase in RACK1 transcriptional activity, protein expression, LPS-induced IL-8, TNF-α production and CD86 expression, highlighting the role of GPER activation in estrogen-induced RACK1 expression. Accordingly, the antagonist G15 blocks RACK1 expression, while agonist G1 and the non-cell permeable BSA-conjugated 17β-estradiol exert a positive action on its expression [150].

#### 3.2.2. RACK1 in Neuronal Context

RACK1 emerges as an interesting player due to its pivotal roles in both physiological and pathological conditions, not only in the immune system but also in the neuronal context [171], where it is highly expressed in the central nervous system (CNS), mainly in neurons, during development and also throughout the adult brain (reviewed in [153]). RACK1 has been described as critical for nervous system development due to its role in regulating axon growth and guidance, point contacts and local translation [172,173,174], although as a scaffolding and ribosomal protein, it displays an important role in proper brain functions, in part due to its interaction with PKC [153]. Indeed, RACK1 deficiency negatively impacts different PKC isoforms’ distribution and localization, subsequently altering specific neural functions [175]. In this regard, impaired PKCβII activation and translocation due to a deficit in RACK1 expression has been found in the post-mortem brains of AD patients and in various aging models [176,177,178,179]. As previously reported, AD is one of the most common neurodegenerative disorders, characterized by the deposition in the brain of fibrillar aggregates of Aβ, derived from APP amyloidogenic proteolytic processing involving the activity of β- and γ-secretases. Conversely, the alternative APP non-amyloidogenic processing, involving the action of α-secretase, cleaves APP within the Aβ region, producing a soluble APP fragment (sAPPα), which exerts neuroprotection, synaptic plasticity, neurite outgrowth and synaptogenesis through the activation of the PI3K/Akt/NF-κB pathway that ultimately positively regulates RACK1 expression and its related signal activity of PKCβII [96,180]. This happens through an NF-κB/c-Rel-related mechanism, as the presence of three consensus NF-κB-responsive elements was detected in the RACK1 promoter region, whose positive regulation has also been found in other contexts, including cancer [181,182]. Moreover, in a neuronal context, the association of RACK1, PKCβII and a variant of the acetylcholinesterase (AChE-R) has been implicated in stress-induced behaviors [153]. Noteworthily, in chronic mild stress (CMS) models, RACK1 was also involved in the spatial and temporal orchestration of the signaling cascade mediated by Brain-Derived Neurotrophic Factor (BDNF) [183,184,185], one of the major mediators of neuroplasticity [186] and adaptation processes in a depressive-like state induced by chronic stress [183,187]. A hallmark of the stress response is the activation of the HPA axis, which has been shown to correlate with RACK1 and ultimately with BDNF expression in a GCs-dependent manner [183,185,187].

##### Glucocorticoids-Mediated RACK1 Transcriptional Regulation in CMS Paradigm

Several intracellular pathways are involved in mediating vulnerability to chronic stress; however, HPA axis activity is pivotal in the maintenance of correct brain processes. The HPA axis is activated in response to stressors, resulting in elevated levels of circulating GCs (cortisol in humans, corticosterone in rodents) and in the functionality of the GRα isoform [80]. In contrast, resilience to chronic stress exposure was correlated with normal GC plasma levels and increased SRSF9 protein associated with overexpression of the GRβ isoform, consistent with the idea that the mechanisms promoting resilience may protect against HPA axis overactivation [188,189]. Accordingly, in the ventral hippocampus of resilient animals (i.e., not developing anhedonic-like behavior), RACK1 gene and protein expression was increased due to the anti-glucocorticoid properties of GRβ, which antagonizes the binding of GRα to the GRE in RACK1 promoter, in a similar mechanism observed in the immune system, leading to the increase of BDNF isoform IV expression. Consequently, animals resilient to chronic stress showed increased RACK1 levels that correlated with a higher BDNF isoform IV expression [183]. Indeed, nuclear RACK1 localizes at *BDNF* promoter IV region and associates with histones H3 and H4, promoting the dissociation of the transcription repressor methyl-CpG-binding protein 2 (MeCP2) and resulting in histone H4 acetylation and promoter-controlled transcription of BDNF exon IV [184]. The BDNF IV has a major role in brain development, dendritic growth, synaptic plasticity, and learning and memory and has been related to neuropsychiatric and neurodegenerative disorders, including depression, anxiety, PD and AD [187].

### 3.3. RACK1 as EDCs Target and Endocrine Disruptor Screening Tool

Regarding the possible identification of RACK1 as an EDC target, we demonstrated that *p*,*p′*DDT and its major metabolite dichlorodiphenyldichloroethylene (*p*,*p′*-DDE), a weak and strong AR antagonist, respectively, reduced RACK1 expression and the innate immune response according to its immune hormonal regulation, thereby decreasing LPS-induced IL-8 and TNF-α release and CD86 expression, previously shown to be dependent on RACK1/PKCβ activation (reviewed in [151]). Three different pesticides from different classes with anti-androgenic properties, the insecticide cypermethrin (CYP), the fungicide vinclozolin (VIN) and the herbicide atrazine (ATR), all of which showed endocrine disruption, were also tested for RACK1 expression and innate immune responses. An anti-androgenic profile was confirmed for CYP due to its effect on proteasome-mediated AR degradation, in line with molecular docking, which excluded CYP as a possible competing anti-androgen [190]. VIN exposure influenced RACK1 expression through a complex mechanism involving nuclear and membrane receptors, specifically AR and GPER. This suggests that VIN and its estrogenic or anti-androgenic metabolites may play a role in modulating immune responses. [190]. Accordingy, VIN caused a slight change in RACK1 expression, while CYP and ATR led to a dose-dependent reduction in RACK1 levels. As a result, there was a decreased release of IL-8 and TNF-α in response to LPS and the reduced expression of the surface markers CD86 and CD54. The ATR-induced RACK1 downregulation was due to the ATR antagonist profile for AR, as reported by docking data, although ATR and its metabolites could exert their anti-androgenic effect by increasing aromatase activity, leading to a decrease in androgen levels [190]. In addition, exposure to these EDCs differently affected the RACK1 levels, pro-inflammation function, natural killer lytic ability and lymphocyte differentiation, with sex-related differences [191]. Hence, RACK1 expression and its associated immune markers can capture both direct and indirect EDC immunotoxicity mechanisms, which do not directly involve steroid hormone nuclear receptors.

DES is a synthetic, non-steroidal estrogen that exerts its activity through its binding to ER. Although no ER-specific consensus sequence was found in the RACK1 promoter, DES exposure significantly increased RACK1 transcription, mRNA and protein levels. This was accompanied by enhanced LPS-induced IL-8 and TNF-α release and increased CD86 expression, indicating that RACK1 is a key target of estrogenic compounds. These effects were shown to be mediated through GPER signaling, which subsequently activates AR, as flutamide completely blocked the DES-induced RACK1 expression. [150]. Consistently, this EDC mechanism of immunotoxicity was also found with the xenoestrogenic mycotoxin zearalenone (ZEA) and the synthetic estrogen ethinylestradiol (EE), which were reported to increase RACK1 expression and correlated with significant LPS-induced cytokine release and CD86. Thus, RACK1 increased levels induced by estrogenic compounds, predisposing cells to an increased response to pro-inflammatory stimuli [150,192]. Nevertheless, BPA, notorious for its hormone-like xenoestrogenic and estrogen-mimicking properties, reduced RACK1 expression and the associated immune response at high concentrations found in workers at BPA manufacturers who reported BPA concentrations of ≤10 μM in their urine samples [12]. In this regard, the literature suggests that BPA’s influence on RACK1 expression is more likely linked to its agonist activity on GRα rather than its antagonistic effect on AR. In addition, a BPA-masked positive effect on RACK1 expression via GPER and related PI3K/Akt/NF-κB activation was detected by using the IκB degradation inhibitor Bay 11-7085, along with the GR antagonist, mifepristone. Indeed, BPA-mediated NF-κB activation is due to the presence of NF-κB/c-Rel binding sites in the RACK1 promoter region [12,162]. Moreover, unlike what has been observed with other EDC estrogen-mimicking properties, although BPA activates GPER, it could still significantly activate the RACK1 promoter upon flutamide exposure, indicating an AR-independent mechanism for a BPA-masked beneficial effect on RACK1 [12]. Due to BPA’s endocrine-disrupting activity, BPA analogs have been developed to replace its use. Considering the growing need to establish bisphenols mechanism(s) affecting the immune system, RACK1 expression and immune-related markers were assessed following BPAF and BPS treatments. Indeed, the inhibition and silencing of specific RACK1 transcriptional modulators can also be employed to unmask the molecular effects of other EDCs. Like BPA, BPAF reduced RACK1 expression and ultimately decreased LPS-induced cytokine release [12,123]. Furthermore, the increased RACK1 expression upon mifepristone exposure suggested additional molecular effects involving GPER-activated AR, as also shown for estrogenic compounds. Consequently, the molecular docking calculations revealed that despite having high structural similarity, BPA and BPAF bind to GPER into two different clefts of GPER-binding sites, supporting BPA’s and BPAF’s different biased agonisms towards GPER [12]. In contrast to BPA and BPAF, BPS induced the expression of RACK1 via AR, which is consistent with its moderately high AR agonist profile predicted by the in silico tools Endocrine Disruptome [193] and VirtualToxLab [12,194].

Finally, diethyl phthalate (DEP) and perfluorooctanesulfonic acid (PFOS) caused a dose-dependent decrease in RACK1 promoter activity, mRNA levels and protein expression, accompanied by reduced IL-8, TNF-α production and CD86 expression. These effects were completely reversed by mifepristone, confirming their GR agonist activity. [192].

This highlights RACK1 as a valuable marker for EDC screening, as it reflects the complex interaction between the transcriptional and non-transcriptional pathways triggered by hormonally active compounds that affect multiple hormonal systems, as summarized in Table 2.

## 4. Involvement of BPA and Its Analogs in INEN: A RACK1-Centered System Toxicology Model Proposal

The evaluation of RACK1 expression and its correlated functions in relation to bisphenol exposure allows us to elucidate new molecular mechanisms in the immuno-neuro-endocrine context. In the neurobehavioral context, RACK1 has been suggested to promote resilience to chronic stress exposure, thus representing a novel target for the treatment of stress-related disorders, including depression [183]. Stress has been described as an environmental factor capable of inducing psychopathology and a depressive phenotype [195] and is also considered a dangerous risk factor for the development of psychiatric disorders. The effect of stress depends on its duration and intensity, as well as the time of life when it first occurs [183]. Studies on animal models showed that stress is associated with the development of depressive behavior and changes in the epigenome [196,197]. Indeed, stress and depression have been associated with epigenetic alterations in the genes involved in mediating resilience and vulnerability to stress, including stress response-related genes [198,199], neurotransmission and the expression of BDNF [200], one of the key mediators of neuroplasticity [201]. As previously stated, BDNF expression is associated with the HPA axis downstream effector RACK1 [162,183,184], whose expression is regulated by a complex hormonal balance in several contexts [1,4,5,202,203]. In stress-related disorders, including depression, a link among GCs, GR, RACK1 and BDNF has been found [183,184,185,200]. The in vivo data report that vulnerability and resilience to the CMS paradigm in rats correlate with a specific activity of the HPA axis. Indeed, the elevation of corticosterone levels observed in response to chronic stress implies a disruption in the negative feedback mechanism that modulates the HPA axis’ function [80]. Moreover, the development of anhedonic-like behavior in stressed animals could be linked to GR splicing isoforms α and β. Specifically, GRα, the classical receptor that binds to GCs, was responsible for a reduction of RACK1 in the ventral hippocampus and, consequently, of RACK1-related demethylation levels in the promoter IV region of the BDNF gene, resulting in a significant decrease in mature BDNF (mBDNF) [183,184]. Indeed, stressful environmental conditions have been shown to cause lasting changes in the methylation of the BDNF promoter IV, leading to reduced BDNF expression and impaired cognitive function. [204,205]. Consistent with toxicological investigations, BPA exposure has been linked to increased BDNF promoter methylation [76], reinforcing the biological basis for the BPA-related behavioral effects observed in epidemiological studies, as summarized in Table 1 and reported later in adulthood. [78]. In line with these considerations, we propose that bisphenol-related depressive symptoms [69,70,71,72,73,74,75,76,77,78,206,207] may be partly linked to their potential effect on RACK1 expression regulation. We suggest that, similarly to the immune system, high BPA and BPAF concentrations could decrease RACK1 levels due to their agonistic properties on GRα, thus affecting BDNF expression [183]. However, these analogs may also act on GRα via an indirect mechanism associated with the HPA axis’ hyperactivation and increased GCs’ release, as previously described. This could be the case with BPS, which increases RACK1 expression in an AR-dependent mechanism, but GCs’ high levels have been shown to counteract the androgen-positive regulation through GRα binding to the RACK1 promoter [161,166]. In the immune context, the GPER-mediated unmasked effect showed that BPA exposure could regulate RACK1 expression through the NF-κB/c-Rel-related mechanism that counteracts GRα [12]. Noteworthily, c-Rel^–/–^ mice (i.e., lacking NF-κB/c-Rel) showed anxiety and depressive-like behavior, suggesting that the characteristics of individuals exposed to typical concentrations of bisphenols may also be essential to predict the acute and/or chronic effects, as observed in the immune context. Moreover, anxiety, depressive-like behavior and apathy are the most common neuropsychiatric symptoms in PD [208]. Indeed, c-Rel^–/–^ mice developed an age-dependent PD-like pathology and phenotype, suggesting that RACK1 expression may be able to capture the bisphenol effects in PD. The relationship between DJ-1 and RACK1 highlights a critical pathway in neuroprotection, particularly relevant to conditions like PD since an altered DJ-1 is associated with early-onset familial PD. DJ-1 functions primarily as a neuroprotective agent, helping to regulate oxidative stress and mitochondrial function. Research suggests that DJ-1 increases the dimerization and stability of RACK1, which is essential for protecting neurons from oxidative stress-induced apoptosis. When DJ-1 is overexpressed, it effectively shields cortical neurons from the oxidative damage caused by agents like hydrogen peroxide (H_2_O_2_). Accordingly, loss of DJ-1 function has been linked to increased oxidative stress, leading to neuronal degeneration and contributing to the pathogenesis of PD. DJ-1 deficiency results in elevated levels of monoamine oxidase B (MAO-B), which is implicated in dopamine degradation and ROS generation, exacerbating mitochondrial dysfunction in neuronal cells. Moreover, the protective effect of DJ-1 is significantly diminished when RACK1 is knocked down, highlighting the importance of their interaction [209,210]. Considering this pathogenic pathway, RACK1 could be an indirect target of bisphenols since BPA significantly increased the expression and oxidation of DJ-1 in the brain of male mice, thus leading to a decrease in RACK1 stabilization that ultimately increased oxidative stress conditions. However, the knockdown of RACK1 could also be due to a direct effect of bisphenols on its expression through a change in the balance of hormonal regulation. Finally, considering that RACK1 levels have also been found to be reduced in the brains of AD patients [176,177,178] and epidemiological studies showed that major depressive disorder (MDD, the stress-related disorder par excellence) could be a risk factor for AD development [211], bisphenol involvement in AD could be partially explained by their direct (agonist effect on GRα) or indirect (HPA axis hyperactivation) effect on RACK1-BDNF-related mechanisms, thus exacerbating anxiety-like behavior and neuroinflammation, similar to the data reported for the prefrontal cortex of adult obese mice [212]. In this context, although BPA and its analogs can exert their detrimental effect on neurons, the literature data report that they can also affect glial cells, such as microglia and astrocytes. Regarding their impact on microglia, exposure to BPA can induce the upregulation of TNF-α and IL-4 mRNA, enhancing inflammatory responses within the brain [108]. Noteworthy, astrocytes also exhibit dosage-dependent effects in response to exposure to BPA and its analogs, including an increased astrocyte differentiation with clear morphological alterations and a reduced astrocyte arborization [213]. Although BPA exposure can activate microglia and astrocytes, triggering the release of inflammatory factors and posing a potential risk for AD’s onset [214], it could be proposed that BPA and its analogs may exert a role on RACK1 expression in astrocytes through GRα and influence the expression of a set of mRNAs encoding inward-rectifying Kir4.1 potassium (K^+^) channels. Indeed, RACK1 has been shown to repress Kir4.1 production in astrocytes and perisynaptic astrocytic processes (PAP) in the hippocampus. Kir4.1 upregulation in the absence of RACK1 enhances Kir4.1-mediated K^+^ currents, leading to alterations in neuronal activity [215] that could be linked to AD. Altogether, these considerations highlight that although differences in the number of bisphenol-interacting receptors or impaired endocrine responses in exposed subjects could be at the basis of the bisphenol effects, the hormonal regulation of RACK1 expression, previously analyzed in the immune system, seems to be able to capture bisphenols and, plausibly, EDCs’ direct or indirect mechanisms involved in the neuronal context and the complexity of bisphenol action on human health (Figure 4).

## 5. Conclusions

An intricate crosstalk exists among the immune, endocrine and nervous systems, marked by dynamic communication and interaction through a constant exchange of signals and information. As previously observed in the case of atrazine [2]—but also for other environmental pollutants [216]—BPA and its analogs also exhibit their effects in INEN through a complex mechanism of endocrine disruption. This involves binding to diverse receptors for endogenous hormones, leading to a complex and nuanced response that varies between tissues and circumstances, depending on the cellular context (i.e., number/type of receptors and interacting proteins). Although BPA’s effects on the endocrine and nervous systems have been extensively studied, consistent data on BPAF’s effects are still lacking, while more attention has been paid to BPF and especially to BPS’s effects, which are also reported in this paper. However, bisphenol’s effects on the immune system need to be further investigated since conflicting data are reported in the literature [117]. This could be ascribed to the lack of epidemiological studies that can support the in vivo and in vitro data, although features of subjects exposed to typical bisphenol concentrations could be essential to predict the acute and/or chronic effects in terms of the BPA-masking effects. Indeed, our in vitro model allows us to better dissect the mechanisms of action of bisphenols on RACK1 expression, thus unmasking the multiple molecular interactions of BPA, BPAF and BPS involving different receptors, both nuclear- and membrane-bound [183]. Moreover, it highlights how differences in the amount of bisphenol-interacting receptors or impaired endocrine responses in exposed subjects may underlie the unmasked effects of bisphenols, which may account for the discrepancy in published data on bisphenols [183]. In this context, RACK1 emerges as a bridge in this intricate communication network and the assessment of its expression could serve to further investigate the toxicity of BPA and its analogs, linking together the immune–neuroendocrine systems. Indeed, RACK1-related transcriptional regulation mechanisms could be detected in both the immune and nervous systems and can be disrupted by BPA and its analogs. In particular, they are able to activate the HPA axis and increase the GC levels that are known to exert their action in both these systems, where the role of RACK1 has been demonstrated and previously discussed. However, our analysis of RACK1 also aims to underline that a systems toxicology approach needs to be considered in analyzing bisphenol’s and EDCs’ effects. Indeed, systems toxicology integrates classical toxicology with the quantitative analysis of large networks of molecular and functional changes that occur across multiple levels of biological organization, providing a deeper understanding [217].

## Figures and Tables

**Figure 1 jox-15-00013-f001:**
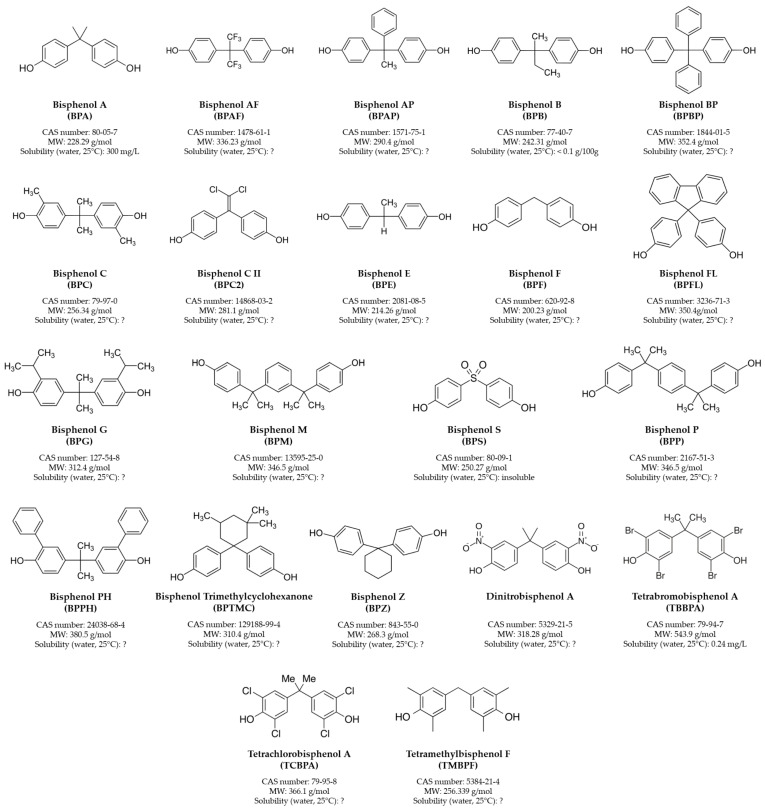
BPA and its major analogs. Reported data were retrieved from PubChem (https://pubchem.ncbi.nlm.nih.gov/, accessed on 18 October 2024). MW = molecular weight. “?” = Solubility not reported.

**Figure 2 jox-15-00013-f002:**
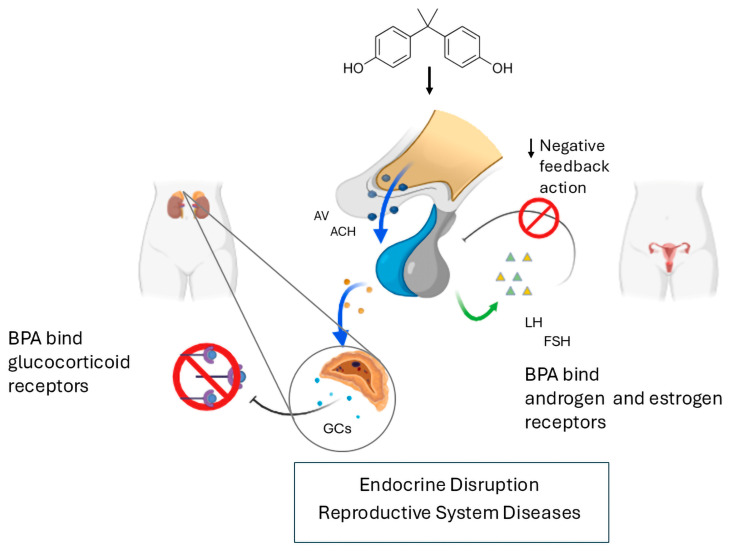
BPA can interfere with estrogen signaling, negatively affecting the release of follicle-stimulating hormone (FSH) and luteinizing hormone (LH). By inhibiting estrogen binding at the pituitary level, BPA can lead to elevated levels of FSH and LH, contributing to reproductive disorders such as polycystic ovary syndrome. In addition, BPA acts as an anti-androgen by binding to and altering the function of androgen and glucocorticoid receptors.

**Figure 3 jox-15-00013-f003:**
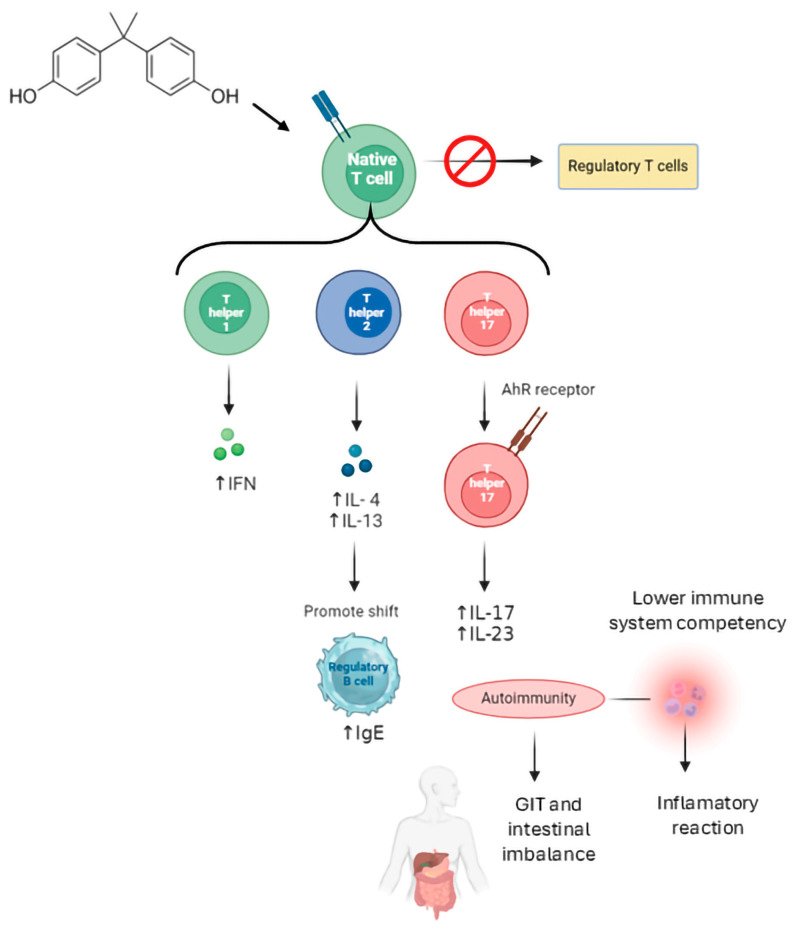
BPA may potentiate autoimmunity by activating Th1, Th2 and Th17 cells. Aryl hydrocarbon receptors (AhRs) play a key role in modulating immune responses, leading to the production of Th17 cells, which are critical in several autoimmune diseases.

**Figure 4 jox-15-00013-f004:**
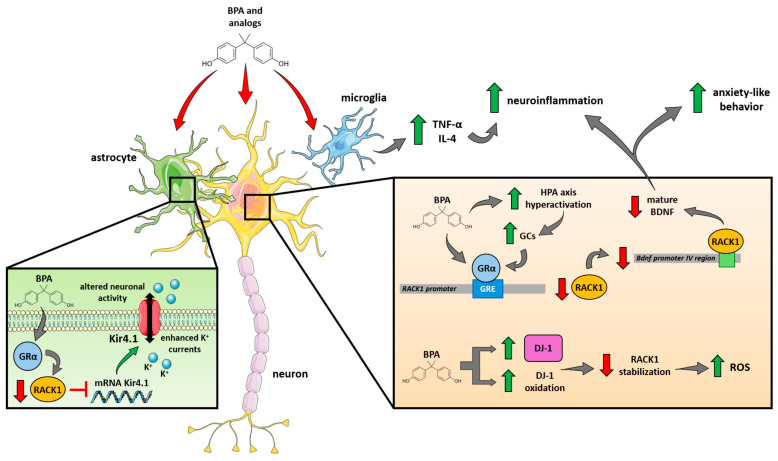
Toxicology model proposal with RACK1 as a central hub to explain bisphenol-mediated adverse effects on INEN. At the neuronal level, BPA can directly (through GRα agonism) and indirectly (through an increased GCs release due to HPA axis hyperactivation) induce RACK1 downregulation, leading to a reduced production of mature BDNF linked to anxiety-like behavior and increased neuroinflammation; in parallel, BPA-induced DJ-1’s increased expression and oxidation contribute to reducing RACK1 stabilization, leading to reduced ROS scavenging and increased oxidative stress. In the microglia, BPA stimulates the production and release of TNF-α and IL-4, contributing to and exacerbating neuroinflammation. Finally, in astrocytes, BPA could activate GRα and reduce RACK1 expression; decreased levels of RACK1, which normally contributes to downregulate Kir4.1 mRNA, lead to an increased Kir4.1 expression, resulting in enhanced K^+^ currents and altered neuronal activity (see text for details).

**Table 1 jox-15-00013-t001:** Epidemiological studies on parental urinary BPA concentrations and emotional neurobehavioral problems in children.

Studied Population	Mean Urinary [BPA]	Observed Effect	Ref.
Pregnant women; 87 boys, 111 girls (3–5 y.o.); CCCEH-NYC prospective cohort (1998–2003).	1.96 µg/L (maternal) 3.94 µg/L (3 y.o.)	BPA prenatal exposure is significantly correlated with emotionally reactive and aggressive behavior only in male offspring.	[68]
601 pregnant women; their 292 children (5 y.o.); Salinas Valley (1999–2000)	1.1 µg/L (maternal) 2.5 µg/L (5 y. o.)	Prenatal urinary [BPA] is correlated with increased anxiety and depression in boys at age 7. Childhood urinary [BPA] associated with hyperactivity behavior in boys and girls at age 7.	[54]
300 children (9–11 y.o.); INMA Granada birth cohort (2011–2012).	4.76 μg/L (children)	BPA exposure in children is significantly associated with somatic complaints, cognitive problems and significant social difficulties.	[69]
2000 women (first trimester); 399 boys, 413 girls (2.8–4.2 years); MIREC, Canada (2008–2011).	0.8 ng/mL (maternal)	BPA exposure is correlated with higher levels of internalizing and somatizing behaviors in boys.	[70]
228 mother–child pairs (127 girls, 101 boys; 8 y.o.); HOME study (March 2003–January 2006)	2.1 ng/mL (prenatal) 1.6 ng/mL (8 y.o.)	BPA prenatal exposure is correlated with higher levels of externalizing behaviors only in girls; BPA exposure in 8 y.o. was correlated with higher levels of externalizing behavior only in boys.	[71]
1225 pregnant women (12–16 weeks); 475 2 y.o. children, 644 4 y.o. children; Shanghai, China	0.31 μg/L (12–16 weeks pregnant women) 0.12 μg/L (4 y.o.)	Maternal urinary [BPA] is associated with elevated risk of emotional reactivity, problematic behavior, anxiety, depression and internalizing problems in male offspring.	[72]
Pregnant women and their children (134 girls, 116 boys; 1–7 y. o.) Pland REPRO_PL birth cohort (2007–2019)	1.9 μg/L (children)	BPA exposure is positively correlated with emotional symptoms; no correlation with cognitive and psychomotor development.	[73]
394 mother–child pairs (2 y.o.) (2009–2012);	1.7 μg BPA/g creatinine (maternal)	Maternal BPA exposure is negatively correlated with social-emotional scores in boys.	[74]
158 boys, 154 girls (2–4 y.o.) APrON study (2009–2012)	1.22 ng/mL (prenatal) 0.93 ng/mL (postnatal)	BPA induced domains of inhibitory self-control and emergent metacognition decline in female offspring.	[75]
668 mother–son pairs (130 boys, 9–11 y.o.) random sample; INMA Project	5.41 μg/g creatinine	Positive correlation during adolescence with cognitive impairments and somatic complaints; BPA exposure correlated with increased BDNF DNA methylation.	[76]

Abbreviations list: BPA concentration ([BPA]); Columbia Center for Children’s Environmental Health (CCCEH); Maternal–Infant Research on Environmental Chemicals (MIREC); New York City (NYC).

**Table 2 jox-15-00013-t002:** Studies employing RACK1 as an in vitro EDC screening tool for their immunotoxicity.

EDC	Effect on RACK1	Mechanism of Action	Models	Concentration	Ref.
BPA	Downregulated (Upregulated, unmasked)	GRα agonist GPER-activated PI3K/Akt/NF-κB signaling	THP-1 PBMCs	0.001–10 μM	[12,123]
BPAF	Downregulated (Upregulated, unmasked)	GRα agonist GPER-activated PI3K/Akt/AR signaling	THP-1 PBMCs	0.001–10 μM	
BPS	Upregulated	AR agonist	THP-1 PBMCs	0.001–10 μM	
DES	Upregulated	GPER-activated PI3K/Akt/AR signaling	THP-1 PBMCs	0.002–20 μM	[150]
ZEA	Upregulated	GPER-activated PI3K/Akt/AR signaling	THP-1 PBMCs	0.001–10 μM	
EE	Upregulated	GPER-activated PI3K/Akt/AR signaling	THP-1 PBMCs	0.001–1 μM	[191,192]
PFOS	Downregulated	GRα agonist	THP-1 PBMCs	0.2–20 μM	
DEP	Downregulated	GRα agonist	THP-1 PBMCs	0.001–10 μM	
*p*-*p*′-DDT	Downregulated	AR antagonist (weak)	THP-1	1–1000 nM	[149]
*p*-*p*′-DDE	Downregulated	AR antagonist (strong)	THP-1	1–1000 nM	
CYP	Downregulated	Reduced AR levels and transcriptional activity (increased SMRT and NCoR recruitment, ARA55 and ARA70 inhibition); IL-6/AR pathway antagonist (anti-androgenic, indirect)	THP-1 PBMCs	0.001–10 μM	[190,191]
VIN	Upregulated (early) Downregulated (late) (promoter only)	GPER-activated signaling (early) AR antagonist (late)	THP-1 PBMCs	0.001–10 μM	
ATR	Downregulated	AR antagonist (direct) Increased aromatase activity (anti-androgenic, indirect)	THP-1 PBMCs	0.001–10 μM	

Abbreviations: Androgen receptor (AR); Androgen receptor-associated protein 55 (*ARA55*); Androgen receptor-associated protein 70 (*ARA70*); Atrazine (ATR); Bisphenol A (BPA); Bisphenol AF (BPAF); Cypermethrin (CYP); Dichlorodiphenyltrichloroethane (*p*,*p*′-DTT); Dichlorodiphenyldichloroethylene (*p*,*p*′-DDE); Diethyl phthalate (DEP); Diethylstilbestrol (DES); Ethinylestradiol (EE); G-protein estrogen receptor (GPER); Glucocorticoid receptor alpha (GRα); Interleukin 6 (IL-6); Nuclear Receptor Corepressor (NCoR); Peripheral blood mononuclear cells (PBMCs); Perfluorooctanesulfonic acid (PFOS); Phosphoinositl-3-kinase (PI3K); Silencing Mediator for Thyroid Hormone Receptors (SMRT); Vinclozolin (VIN); Zearalenone (ZEA).

## Data Availability

No new data were created or analyzed in this study.

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
