# Peer review of "Endocrine Disrupting Toxicity of Bisphenol A and Its Analogs: Implications in the Neuro-Immune Milieu"

_jox, 2025, doi:10.3390/jox15010013_

Round 1
Reviewer 1 Report
Comments and Suggestions for Authors
Line 41: Part of the sentence “that are not directly the target” would imply that there are organisms on which these substances act directly? It was probably meant something like DDT, as mentioned later in the manuscript, but here at the beginning it could be related to BPA.
Line 32 and throughout the manuscript: Please write “in vitro” and “in vivo” in italics
Tables and figures: Please indicate the source of the tables and figures if they were not prepared by the authors of this manuscript
Table 1: In some parts of this table some information is missing, e.g. the indication of women/children next to the BPA concentration in the columns “Mean urine values” and “Observed effect”; next to the reference (50) the age 7 is indicated in the third column, while the study refers to age 5
Author Response
Comment 1: Line 41: Part of the sentence “that are not directly the target” would imply that there are organisms on which these substances act directly? It was probably meant something like DDT, as mentioned later in the manuscript, but here at the beginning it could be related to BPA.
Response 1: We apologize to the reviewer for the errata. We originally intended to address EDCs’ ability to also affect subpopulations that may have not been directly exposed to the inquired substance. Hence, to avoid any misunderstanding, we rephrased the sentence as follows (lines 37-39): “According to the European Commission, a substance is defined as an endocrine disruptor if it causes an adverse effect either in an organism or its progeny, or (sub)populations”.
Comment 2: Line 32 and throughout the manuscript: Please write “in vitro” and “in vivo” in italics
Response 2: We modified the text as requested.
Comment 3: Tables and figures: Please indicate the source of the tables and figures if they were not prepared by the authors of this manuscript
Response 3: In light of the reviewer’s observation, we would like to point out that all tables and figures were specifically made by the authors for this original manuscript and no third party-produced images or tables were adapted for this review article. In particular, we employed ChemDraw to generate Figure 1 and SMART – Servier Medical ART for Figures 2-4. To clarify this point, we modified the Acknowledgements section as follows (lines 859-861): “The authors would like to thank ChemDraw used to prepare Figure 1 and SMART—Servier Medical ART (https://smart.servier.com/, accessed on 22 October 2024) used for Figure 2-4”.
Comment 4: Table 1: In some parts of this table some information is missing, e.g. the indication of women/children next to the BPA concentration in the columns “Mean urine values” and “Observed effect”; next to the reference (50) the age 7 is indicated in the third column, while the study refers to age 5
Response 4: We are thankful to the reviewer for pointing out this issue. As requested, we included all missing information in Table 1. Moreover, in view of the reference 50 given in Table 1, we would like to clarify that the BPA concentrations at the age of 5 years refer to those reported at the beginning of the study, whereas the effects were observed at the age of 7 years in order to assess the existence of a correlation between urinary BPA concentrations and the development of behavioural problems. Therefore, the study reported the final effect at 7 years of age.
Reviewer 2 Report
Comments and Suggestions for Authors
Abstract:
Please replace (line 17) “and their proven or hypothesized implications in multiple pathological conditions” with “the potential levels of long-term exposure with implications in multiple pathological conditions”.
Please replace (line 20) “whose widespread use” with “whose Tolerable Daily Intake (TDI), was re-evaluated in 2023 by the European Food Safety Authority (EFSA).
Line 27 please rerplace “neuronal and immune systems” with “immune and nervous system”.
Introduction section:
line 36: please replace “block” with “antagonize”.
Line 37 added “and thus the endocrine system” after body’s homones, and remove “due to their potential to interfere with the endocrine system”.
Lines 41-44: the sentence is too long and not clear “These adverse effects, which include changes in the morphology, physiology, growth, development, reproduction, or lifespan of an organism, system, or population/subpopulation, can lead to negative changes in functional capacity, in the ability to compensate for additional stress, or in increased susceptibility to other influences”. What do you mean with subpopulation?
Line 49 replace “immune, nervous, and endocrine system” with “endocrine, immune, and nervous systems”.
Paragraph 2
Line 80 Do the authors refer to all bisphenols or just to BPA, as mentioned at the beginning of the sentence?
Line 84: What do the authors mean with “mainly occurs”? The authors should consider other workers including the cashiers exposed via dermal penetration to BPA through thermal receipts (see for instance Lv Y et al., Environ Int. 2017) and other workers, e.g. employees of sewage-pipe relining companies and floor-coating companies (Porras SP et al., Toxicol Lett. 2024).
Lines 98-100: please consider also breast milk, cow milk and dairy products (e.g. Mercogliano, Santonicola, Food Chem Toxicol. 2018).
Line 106 please added the definition of TDI, as the threshold for daily human exposure to a chemical pollutant considered safe for consumer health.
Line 128: rephrase the tile 2.2.1: Effects of BPA and its analogs on neurodevelopment and behavior.
Line 133 replace “to” with “of”.
Line 141: please replace “histological structures” with “tissues”.
Please in figure 2 and 3 remove the immages of plastics and add the structures of BPA, according to figure legends, since plastic picture can be confusing.
Line 235 correct the typos Neurogenerative into neurodegenerative.
Line 335: replace “motile” with “mobile”.
Replace “for up to 96 h” with “over 96 hours” and remove parentheses around "a widely used cell line" and replace with “a widely used model for primary human macrophages”.
Please rephrase: “Literature data report that in prenatal and adult exposed mice, BPA promotes…” with “Literature data report that in mice exposed during prenatal stages or adulthood, BPA promotes…”
Line 438: rephrase the title of 2.2.4 paragraph: “BPA and its analogs on other related diseases” into “BPA and its analogs on other immunoendocrine-related diseases”.
At the end of paragraph 2.2.4 please cite the study showing that BPA exposure represents an additional risk factor for the progression of fatty liver diseases strictly related to the cross-talk between inflammation, oxidative stress and immune-metabolic impairment associated to obesity in mice (Pirozzi et al., Antioxidants (Basel). 2020 Nov 30;9(12):1201. doi: 10.3390/antiox9121201. PMID: 33265944).
In order to introduce the link between the EDCs and RACK1 write a little introduction before 3.1 paragraph. A suggestion could be to move lines 596-600 “Among the best known and studied EDCs, the insecticide dichlorodiphenyltrichloroethane (p,p'-DTT) and the synthetic non-steroidal estrogen diethylstilbestrol (DES) were central proof-of-concept substances to the discovery of endocrine disruption and were used in the initial investigation to support RACK1 as a possible target of EDCs and their immunotoxicity [5,149,163,164]” before 3.1 paragraph.
Line 581: what is the meaning of “resilient” animals? Please add “to chronic stress exposure”.
Line 600: to better underline the value of the author’s data, the sentence “In this regard…” could be modified as follows “Regarding the possible identification of RACK1 as EDCs target, we demonstrate that…”
Conclusions
Line 808: The relationship between EDCs and endocrine disruption cannot be reduced to a single environmental pollutant, eg. atrazine. Therefore replace “atrazine” with “other environmental pollutants” (ref. Yilmaz et al. Rev Endocr Metab Disord. 2020 Mar;21(1):127-147. doi: 10.1007/s11154-019-09521-z. PMID: 31792807).
Author Response
We would like to thank the reviewer for its time and effort and for the thorough revision process. We modified the text as requested and implemented all the suggested literature.
Comment: Lines 41-44: the sentence is too long and not clear “These adverse effects, which include changes in the morphology, physiology, growth, development, reproduction, or lifespan of an organism, system, or population/subpopulation, can lead to negative changes in functional capacity, in the ability to compensate for additional stress, or in increased susceptibility to other influences”. What do you mean with subpopulation?
Response: We agree with the reviewer’s observation and rephrased the sentence as follows (lines 39-42): “These adverse effects, which include alterations in the morphology, physiology, growth, development, reproduction, or lifespan, can lead to negative changes in functional capacity, in the ability to compensate for additional stress, or to an increased susceptibility to other influences”.
Comment: Line 80 Do the authors refer to all bisphenols or just to BPA, as mentioned at the beginning of the sentence?
Response: We apologize for the overlooked error. As the reviewer correctly stated, we intended to refer to all bisphenols. Hence, we modified the text as follows (lines 74-78): “Hence, the toxicity of BPA and its analogs is related to their interaction with hormone receptors on specific cell types whose expression is modulated according to age, sex, endocrine status, and brain region. Indeed, this highlights that exposure to bisphenols could be associated with different adverse health effects in humans (reviewed in [15,16])”.
Comment: Line 84: What do the authors mean with “mainly occurs”? The authors should consider other workers including the cashiers exposed via dermal penetration to BPA through thermal receipts (see for instance Lv Y et al., Environ Int. 2017) and other workers, e.g. employees of sewage-pipe relining companies and floor-coating companies (Porras SP et al., Toxicol Lett. 2024).
Response: We thank the reviewer for the suggestion and we modified the text accordingly (lines 81-87): “Occupational exposure occurs in workers involved in the synthesis of BPA and its derivatives (i.e. polycarbonate, epoxy resins and polyvinyl chloride), but also in cashiers exposed to BPA via dermal penetration through thermal receipts [18] and other workers, e.g. employees of sewage-pipe relining companies and floor-coating companies [19]. On the other hand, while environmental exposure is due to its use in thermal paper recycling and related industries, resulting in contamination of the atmosphere, soil and aquatic systems [20]”.
Comment: Lines 98-100: please consider also breast milk, cow milk and dairy products (e.g. Mercogliano, Santonicola, Food Chem Toxicol. 2018).
Response: We agree with the reviewer and modified the text accordingly (lines 99-100): “In addition, breast milk, cow milk and dairy products represent additional notable sources for BPA oral exposure [23].”.
Comment: Line 106 please added the definition of TDI, as the threshold for daily human exposure to a chemical pollutant considered safe for consumer health.
Response: We thank the reviewer for the suggestion. Accordingly, we modified the sentence as follows (lines 105-108): “The European Union Food Safety Authority (EFSA) set a tolerable daily intake (TDI) – defined as the threshold for daily human exposure to a chemical pollutant considered safe for consumer health – for BPA of 0.2 ng/kg body weight (bw)/day for human health over a lifetime of chronic exposure”
Comment: Please in figure 2 and 3 remove the images of plastics and add the structures of BPA, according to figure legends, since plastic picture can be confusing.
Response: We agree with the reviewer’s observation and modified the text and the figures accordingly.
Comment: At the end of paragraph 2.2.4 please cite the study showing that BPA exposure represents an additional risk factor for the progression of fatty liver diseases strictly related to the cross-talk between inflammation, oxidative stress and immune-metabolic impairment associated to obesity in mice (Pirozzi et al., Antioxidants (Basel). 2020 Nov 30;9(12):1201. doi: 10.3390/antiox9121201. PMID: 33265944).
Response: The reviewer is right in pointing out these aspects of BPA exposure and obesity and we agree that their insertion is necessary to comprehensively report BPA features in this context. Accordingly, we modified the text as follows (lines 451-456): “In this regard, BPA exposure has been described as an additional risk factor for the progression of fatty liver diseases strictly related to the cross-talk between inflammation, oxidative stress and immune-metabolic impairment associated with obesity in mice [142]. Besides BPA, various population-based studies revealed that BPA substitutes (BPAF, BPF and BPS) may act as obesogens at pathophysiological level (reviewed [143]).”.
Comment: In order to introduce the link between the EDCs and RACK1 write a little introduction before 3.1 paragraph. A suggestion could be to move lines 596-600 “Among the best known and studied EDCs, the insecticide dichlorodiphenyltrichloroethane (p,p'-DTT) and the synthetic non-steroidal estrogen diethylstilbestrol (DES) were central proof-of-concept substances to the discovery of endocrine disruption and were used in the initial investigation to support RACK1 as a possible target of EDCs and their immunotoxicity [5,149,163,164]” before 3.1 paragraph.
Response: We are thankful to the reviewer for the insightful suggestion and we agree that the proposed text modification can better introduce the link between EDCs and RACK1. Accordingly, we moved the indicated lines and modified the manuscript accordingly.
Comment: Line 581: what is the meaning of “resilient” animals? Please add “to chronic stress exposure”.
Response: The reviewer is right in pointing out that in this part of the manuscript a clear definition of resilient (and, thus, vulnerable) animals is lacking. Referring to our previous publication [doi: 10.1016/j.ynstr.2021.100372], sucrose consumption test (extensively employed in CMS protocols) allowed the identification of two different subgroups of rats developing, or not, the anhedonic-like behaviour, thus determining two populations that are vulnerable or resilient, respectively, to chronic stress. Therefore, to make explicit this aspect, we modified the text as follows (lines 598-603): “Accordingly, in the ventral hippocampus of resilient animals (i.e., not developing anhedonic-like behavior), RACK1 gene and protein expression was increased due to the anti-glucocorticoid properties of GRβ, which antagonizes the binding of GRα to the GRE in RACK1 promoter, in a similar mechanism observed in the immune system, leading to the increase of BDNF isoform IV expression”. In addition, we also added “to chronic stress exposure” (line 595) as suggested by the reviewer.
Comment: Line 600: to better underline the value of the author’s data, the sentence “In this regard…” could be modified as follows “Regarding the possible identification of RACK1 as EDCs target, we demonstrate that …”
Response: We thank the reviewer for the suggestion and we rephrased the sentence as indicated (lines 615-616)
Comment: Line 808: The relationship between EDCs and endocrine disruption cannot be reduced to a single environmental pollutant, eg. atrazine. Therefore replace “atrazine” with “other environmental pollutants” (ref. Yilmaz et al. Rev Endocr Metab Disord. 2020 Mar;21(1):127-147. doi: 10.1007/s11154-019-09521-z. PMID: 31792807).
Response: We apologize to the reviewer for the misunderstanding. We are aware that endocrine disruption is a complex mechanism whose effects are not limited to a single tissue or environment pollutant. However, in the indicated portion of text, we originally intended to highlight the similarities of the detrimental effects of endocrine disruption, specifically on INEN, observed for BPA/BPA analogs and atrazine, on which we focused our attention in a previous publication [doi: 10.3389/ftox.2021.649024] and that served as proof-of-concept EDC to stress how the nervous system (and, thus, INEN) is an underrepresented EDC target and the importance to further investigate EDC toxicity in this context. Nevertheless, to meet the reviewer’s request and better cover the state of the art on this matter, we modified the text as follows (lines 818-820): “As previously observed in the case of atrazine [2] – but also for other environmental pollutants [211]– BPA and its analogs also exhibit their effects in INEN through a complex mechanism of endocrine disruption”.
Reviewer 3 Report
Comments and Suggestions for Authors
Review of the paper « Endocrine disrupting toxicity of bisphenol A and its analogs: implications in the neuro-immune milieu » by Buoso et al.
General comment:
This is an interesting review focusing on Bisphenols for their neuro and immune effects and mechanisms of action. Indeed, the neuronal and immune systems, key targets of hormonal signaling, are emerging as critical players in endocrine disruption. This review also highlight Receptor for Activated C Kinase 1 (RACK1) as a target for EDC (direct or not). It is well written and illustrated and clear.
Introduction:
L44: suggestion susceptibility à sensitivity
L45-47 : Can you reword the following sentence for clarity or make it two sentences : “Furthermore, a substance is classified as an EDC when its mode of action is endocrine, that is, whether it has the ability to change the functions of the endocrine system and, therefore, the adverse effect is a consequence of this endocrine disruption [1].”
L53: the aim of a review is more to summarize what is reported on a specific topic or to suggest a mechanism of action than to elucidate anything
L85-87: Why do you claim that the environmental contamination is solely due to the thermal paper use of bisphenol :”…while environmental exposure is due to its use in thermal paper recycling and related industries, resulting in contamination of the atmosphere, soil and aquatic systems [18].”
As it is widely used in a large variety of everyday life products, I would think that the environmental contamination corresponds to the sum of all of that and not solely to the thermal paper use.
As indicated line 96.
I suggest to remove the sentence L85-87 indicated above.
L100 Data corresponding to the % of exposed people or of plasmatic concentration detected are missing, especially for the 4 bisphenol of interest BPA/BPAF/BPS/BPF
Figure 1 and its legend are misplaced in the pdf I have
L129-140: misplaced. Paragrapgh corresponding to absorption/metabolization, should be placed elsewhere and not in the section corresponding to “BPA and its analogs on neurodevelopment and behavior “
L141 : I haven’t seen the following review in your health effect section
McDonough CM, Xu HS, Guo TL. Toxicity of bisphenol analogues on the reproductive, nervous, and immune systems, and their relationships to gut microbiome and metabolism: insights from a multi-species comparison. Crit Rev Toxicol. 2021 Apr;51(4):283-300. doi: 10.1080/10408444.2021.1908224. Epub 2021 May 5. PMID: 33949917.
L143 please reword. I guess the brains were collected during the autopsy but I doubt that the BPA accumulation was evidenced during the autopsy. Just remove “during the autopsy”
L143-146 : also misplaced, should be with the exposure paragraph
L154: what do you mean by “sublethal concentrations” ? do you mean really high dose? What concentration? Is it compatible with human exposure?
L173 suggestion: Recently, BPA and BPS potential influence on child behavior at 2 years was assessed using the Child Behavior Checklist (CBCL).
L180 remove “However” (because no link and no opposition with the previous sentence).
L183 suggestion : “Indeed, in elderly men…”
Figure 2 and its legend are misplaced
While the paragraph (L190-209) is detailed on the HPA it doesn’t mention effect on reproductive system, contrary to the figure 2 and its legend (L210-214). This should therefore be also mentioned in the text.
Figure2: I don’t understand what is depicted behind the red symbol. You should indicate what is represented
L239, 242 and elsewhere “following bisphenol exposure” (bisphenol is singular as it is an adjective, if you want to keep it plural you have to modify the sentence à following exposure to bisphenols)
L248 higher meaning? 2-fold? 10%? 1%?
2.2.2 the title is a bit misleading : absence of information on BPS/BPF
L236-291 à BPA
L291-295 à BPF/BPS
L296-317 à BPA
L317-320à BPF/BPS
Given the title I would have expected more data on the analogs, at least the scarcity of data should be mentionned
L328 BPA analog effects (same reason as mentioned above)
L368 cytokines (remove”-“)
Figure 3 : indicate bisphenols on the waste in pink at the top.
Comment : mineral water bottles should contain more Polyethylene terephthalate (PET) than bisphenols, not sure it is the most relevant representation of bisphenol waste as it might be misleading for readers
Separate Fig3 legend from the following paragraph for clarity
L401 : “environmentally relevant concentrations” but as mentioned above, exposure data are missing from the introduction section
The environementally relevant concentrations only correspond to the first 2 sentences. The rest of the paragraph only mention effect observed at supraenvironmental level of exposure. This should be indicated as to not mislead the reader.
L430 low doses (give the range)
L434: remove “that” (one of the 2 that)
Section 2.2.4 suggestion: I actually would have started with this, indicating several health effects that won’t be detailed in this review (but it is those health effect and the correspondinf mechanisms of action that lead to the classification of bisphenol A as an endocrine disruptor) to then report in details neuro and immuno effects of bisphenols
L476 cytokine release
L529 monocyte activation
Section 3.3 : no discrepancies reported that wouldn’t support your hypothesis?
L691 bisphenol exposure
L740 bisphenol effects
L762 bisphenol involvement
L785 bisphenol actions
L816 bisphenol effect
L819 bisphenol concentration
Author Response
Comment: This is an interesting review focusing on Bisphenols for their neuro and immune effects and mechanisms of action. Indeed, the neuronal and immune systems, key targets of hormonal signaling, are emerging as critical players in endocrine disruption. This review also highlight Receptor for Activated C Kinase 1 (RACK1) as a target for EDC (direct or not). It is well written and illustrated and clear.
Response: We would like to thank the reviewer for the kind words on the quality of our manuscript and for the upraising comments on our work here presented. All minor changes have been made (even though we did not directly addressed in the form of Comment/Response). We modified the text as requested and implemented all the suggested literature.
Comment: L45-47: Can you reword the following sentence for clarity or make it two sentences: “Furthermore, a substance is classified as an EDC when its mode of action is endocrine, that is, whether it has the ability to change the functions of the endocrine system and, therefore, the adverse effect is a consequence of this endocrine disruption [1].”
Response: As suggested by the reviewer, we rearranged the sentence as follows (lines 42-45): “Furthermore, a substance is classified as an EDC when it has the ability to change the functions of the endocrine system and, therefore, its adverse effects are a consequence of the disruption of endocrine functions [1]”.
Comment: L53: the aim of a review is more to summarize what is reported on a specific topic or to suggest a mechanism of action than to elucidate anything.
Response: We agree with the reviewer and modified the text accordingly (lines 50-53): “Hence, the aim of this review is to underscore the effect of bisphenol A (BPA) and its widely used analogs BPAF, BPF and BPS, in this intricate communication net-work, which depends on the cellular context in terms of receptors and interacting proteins and may therefore differ between tissues and conditions.”
Comment: L85-87: Why do you claim that the environmental contamination is solely due to the thermal paper use of bisphenol: ”…while environmental exposure is due to its use in thermal paper recycling and related industries, resulting in contamination of the atmosphere, soil and aquatic systems [18].” As it is widely used in a large variety of everyday life products, I would think that the environmental contamination corresponds to the sum of all of that and not solely to the thermal paper use. As indicated line 96. I suggest to remove the sentence L85-87 indicated above.
Response: We agree with the reviewer and apologize for the sentence misconstruction. Hence, also to meet other reviewers’ requests, we modified tis part of text as follows (lines 81-87): “Occupational exposure occurs in workers involved in the synthesis of BPA and its derivatives (i.e. polyvinyl chloride, polycarbonate and epoxy resins), but also in cashiers exposed to BPA via dermal penetration through thermal receipts [18] and other workers (e.g. employees of sewage-pipe relining companies and floor-coating companies) [19]. On the other hand, environmental exposure – also as a consequence of its use in thermal paper recycling and related industries – results in contamination of the atmosphere, soil and aquatic systems [20]”.
Comment: L100 Data corresponding to the % of exposed people or of plasmatic concentration detected are missing, especially for the 4 bisphenol of interest BPA/BPAF/BPS/BPF
Response: We thank the reviewer for this observation. We would like to clarify that this sentence serves as a summary of the potential routes of dermal exposure to bisphenols. In order to facilitate a comprehensive and immediate overview for the reader, whilst also ensuring that data has not been redundantly published and extensively discussed in recent research, the decision was taken to reference these studies and indicate them as “reviewed in [23]”.
Comment: Figure 1 and its legend are misplaced in the pdf I have
Response: We apologize for the inconvenience. Considering that in the originally uploaded version of the manuscript Figure 1 and its respective figure legend are correctly placed, this must be an after-upload issue. Nevertheless, we checked the correct Figure 1 placement and guarantee that it appears in its proper position in the revised manuscript.
Comment: L129-140: misplaced. Paragraph corresponding to absorption/metabolization, should be placed elsewhere and not in the section corresponding to “BPA and its analogs on neurodevelopment and behavior “; L143-146: also misplaced, should be with the exposure paragraph
Response: We would like to thank the reviewer for pointing out this issue. We agree with their observation and moved the paragraph accordingly (lines 115-139).
Comment: L141: I haven’t seen the following review in your health effect section: McDonough CM, Xu HS, Guo TL. Toxicity of bisphenol analogues on the reproductive, nervous, and immune systems, and their relationships to gut microbiome and metabolism: insights from a multi-species comparison. Crit Rev Toxicol. 2021 Apr;51(4):283-300. doi: 10.1080/10408444.2021.1908224. Epub 2021 May 5. PMID: 33949917.
Response: We thank the reviewer for the insightful suggestion and we apologize for having overlooked this reference. As suggested, we implemented this article (together with other bibliography modifications suggested by the other reviewers) among the other cited studies.
Comment: L143 please reword. I guess the brains were collected during the autopsy but I doubt that the BPA accumulation was evidenced during the autopsy. Just remove “during the autopsy”
Response: We thank the reviewer for the suggestion. We modified the sentence as follows (lines 127-128): “Indeed, BPA accumulation was reported in post-mortem human brains”.
Comment: L154: what do you mean by “sublethal concentrations”? do you mean really high dose? What concentration? Is it compatible with human exposure?
Response: We apologize to the reviewer for not having better specified this aspect. As for “sublethal concentrations” we referred to the cited studies (10.1016/j.ntt.2024.107348; 10.1016/j.ecoenv.2023.114643), which defined the sublethal concentrations as the 1%, 10% or 100% EC25 of each compound (different for each BPA analog) and calculated based on dose-response curves. Hence, to better clarify this aspect, we modified the sentence as follows (lines 154-158): “Moreover, recent in vivo animal data suggest that BPF and BPS, at sublethal concentrations (1, 10 or 100% their respective EC25), may also pose significant risks to ASD development in humans, highlighting the importance of a comprehensive assessment of developmental neurotoxicity effects for bisphenols [62,63]”.
Comment: Figure 2 and its legend are misplaced; Figure2: I don’t understand what is depicted behind the red symbol. You should indicate what is represented.
Response: We thank the reviewer for the observation. As for other similar comments, we would like to point out that figure misplacement must have occurred after the manuscript file upload. However, we made sure that figures and respective legends are correctly placed in the revised version. In addition, as also suggested by another reviewer, we modified Figure 2 and Figure 3 removing the plastic icon and adding BPA structure.
Comment: While the paragraph (L190-209) is detailed on the HPA it doesn’t mention effect on reproductive system, contrary to the figure 2 and its legend (L210-214). This should therefore be also mentioned in the text.
Response: We thank the reviewer for the observation. We modified the text as follows (lines 209-214):” In this regard, BPA has been reported to interfere with hormonal balance by disrupting estrogen and androgen receptor function, leading to altered levels of follicle stimulating hormone (FSH) and luteinizing hormone (LH). This imbalance can contribute to reproductive disorders, including polycystic ovary syndrome (Figure 2). Finally, other downstream physiological consequences include also metabolic and cardiovascular diseases [83-86]”.
Comment: L248 higher meaning? 2-fold? 10%? 1%?
Response: We thank the reviewer for the clarification, we have amended it as follows (lines 250-253): “Epidemiological studies report that BPA concentrations detected in the liver and adipose tissue of patients with AD were, respectively, 4.24-fold and 6.76-fold higher than those of the age-matched control group [90]”.
Comment: 2.2.2 the title is a bit misleading: absence of information on BPS/BPF. L236-291 à BPA; L291-295 à BPF/BPS; L296-317 à BPA; L317-320à BPF/BPS Given the title I would have expected more data on the analogs, at least the scarcity of data should be mentioned.
Response: We agree with the reviewer on the scarcity of data regarding the correlation between BPA analogs and the development of neurodegenerative disorders. Hence, to meet the reviewer’s request, we modified the text as follows (lines 299-301) “Nevertheless, the scarcity of experimental evidence on the correlation between BPA analogs and the development of neurodegenerative diseases like AD surely warrants further investigations to address these aspects”. However, since some observations on BPF and BPS possible involvement in this context were actually reported, we decided to keep the original title to comprehensively address the topic.
Comment: L328 BPA analog effects (same reason as mentioned above)
Response: We modified the text accordingly (lines 330-332) “In this context, despite numerous original studies and reviews on BPA immunotoxicity, only a low number of studies investigated the effects of BPA analogs on the immune system”.
Comment: Figure 3: indicate bisphenols on the waste in pink at the top. Comment: mineral water bottles should contain more Polyethylene terephthalate (PET) than bisphenols, not sure it is the most relevant representation of bisphenol waste as it might be misleading for readers
Response: We thank the reviewer for the suggestion. As also indicated by another reviewer, we modified Figure 2 and Figure 3 removing the plastic icon and adding BPA structure.
Comment: Separate Fig3 legend from the following paragraph for clarity
Response: We thank the reviewer for pointing out the issue. As for Figure 1 misplacement, in our originally submitted manuscript, figures and figure legends were correctly placed. In the present revised version, we ensured that figures and their respective legends were correctly positioned.
Comment: L401: “environmentally relevant concentrations” but as mentioned above, exposure data are missing from the introduction section. The environmentally relevant concentrations only correspond to the first 2 sentences. The rest of the paragraph only mention effect observed at supra-environmental level of exposure. This should be indicated as to not mislead the reader.
Response: We thank the reviewer for the observation. We modified the text as follows (lines 408-413):” In addition to BPA, the effects of exposure to environmentally relevant concentrations (i.e. 0.05 nM) of BPF and BPS were analyzed in mouse spleen T lymphocytes. Exposure to 0.05 nM BPA or BPAF significantly increased IL-17 levels, whereas BPS had no effect [128]. In contrast, on human T cells, only the highest concentration of bisphenols, representing the supra-environmental level of exposure (50 µM BPA, BPF or BPS), significantly reduced the secretion of IL-17 and IL-22 [128]”.
Comment: L430 low doses (give the range)
Response: We apologize for not having clarified this aspect before. We checked the original manuscript to retrieve the requested range and modified the text as follows (lines 437-439): “Specifically, low doses (0.1-10 nM) of BPA can enhance cell proliferation while simultaneously decreasing the secretion of anti-inflammatory cytokines, particularly in activated PBMCs”
Comment: Section 2.2.4 suggestion: I actually would have started with this, indicating several health effects that won’t be detailed in this review (but it is those health effect and the corresponding mechanisms of action that lead to the classification of bisphenol A as an endocrine disruptor) to then report in details neuro and immuno effects of bisphenols.
Response: We thank the reviewer for the suggestion. However, in an attempt to meet another reviewer’s request, we modified the 2.2.4 to “BPA and its analogs on other immunoendocrine-related diseases” section also to implement another study. Hence, considering that the previous section referred to BPA effect on the immune system, we deemed correct to leave the aforementioned section after the one addressing BPA immune effects.
Comment: Section 3.3: no discrepancies reported that wouldn’t support your hypothesis?
Response: In light of the reviewer’s observation, we wish to clarify that the results reported here are consistent with the hypothesis that RACK1 serves as a reliable screening tool for endocrine disrupting chemicals (EDCs). No significant discrepancies were also found in literature that would undermine its proposed role in capturing the immunotoxicity mechanisms of endocrine disrupting compounds.